# Instrumental Activities of Daily Living—A Good Tool to Prospectively Assess Disability after a Second Contralateral Hip Fracture?

**DOI:** 10.3390/geriatrics5040067

**Published:** 2020-09-29

**Authors:** Emilija Dubljanin Raspopovic, Ljiljana Marković Denić, Sanja Tomanović Vujadinović, Marko Kadija, Una Nedeljković, Nela Ilić, Darko Milovanović

**Affiliations:** 1Clinic for Physical Medicine and Rehabilitation, Clinical Center Serbia Faculty of Medicine, University of Belgrade, 11000 Belgrade, Serbia; drsanjatv@gmail.com (S.T.V.); kadija.marko@gmail.com (M.K.); unaned@gmail.com (U.N.); nelavilic@gmail.com (N.I.); darkomil@doctor.com (D.M.); 2Institute of Epidemiology, Faculty of Medicine, University of Belgrade, 11000 Belgrade, Serbia; ljm.denic@gmail.com

**Keywords:** hip fractures, disability, rehabilitation, mortality, dementia, secondary hip fracture

## Abstract

The aim of this study was to determine the outcome for patients who sustain a second hip fracture compared with those who sustain a first fracture, and to define the optimal measure to evaluate functional outcome after second hip fracture. Methods: 343 patients with acute hip fractures who presented during a 12 month period were included in the study. Patients with a first (318 patients, 78.10 +/− 7.53 years) and second (25 patients, 78.96 +/− 6.02) hip fracture were compared regarding all baseline variables. Regression analysis was also performed to assess the independent relationship between the presence of a second hip fracture and observed outcome variables at discharge (physical disability, complications, length of stay, and mortality) and one-year after surgery (physical disability and mortality). Results: Disability when performing instrumentalized activities of daily living (IADL) at one-year follow-up is independently related to the presence of a second hip fracture. There were no other statistically significant relationships between the presence of a second hip fracture and other observed outcome variables. Conclusions: Patients with a second hip fracture showed worse functional outcome at one-year follow-up when measured with the IADL scale. No increased short-nor long-term mortality rates were found in patients with a secondary hip fracture. IADL is a good tool to assess disability after a second hip fracture and could be thus a more reliable outcome measure when investigating differences in functional recovery in patients with a second hip fracture compared to conventionally used ADL scales.

## 1. Introduction

The risk of sustaining a second hip fracture is substantially increased in surviving post-fracture hip patients [1]. The incidence of second hip fractures ranges from 6% to 12% in different studies. Several factors such as female gender, older age, dementia, Parkinson disease, history of falls, low vision, osteoporosis, cardio-respiratory disease, and instutionalization have been found to increase the risk of a second hip fracture [2,3,4]. It would be reasonable to assume that patients who sustain a second hip fracture are predisposed to worse outcome in terms of physical disability and increased mortality. However, a limited number of studies reveal controversial results regarding this topic [2,5,6,7,8]. The majority of studies report worse functional outcomes [6,7,8,9] and higher one-year mortality [2,3,10] in patients with a second hip fracture, while two authors failed to show worse functional results in this patient group [1,2].

Understanding the epidemiologic characteristics and outcome of patients who sustain a second contralateral hip fracture may lead to improved prevention and rehabilitation strategies.

The aim of this study was to determine the outcome for patients who sustain a second hip fracture compared with those who sustain a first fracture, and to define the optimal measure to evaluate functional outcome after second hip fracture.

## 2. Materials and Methods

### 2.1. Participants

We examined 384 patients who presented with acute first and second hip fractures to a university-associated orthopedic hospital during a 12 month period. This teaching hospital has a catchment population of approximately 1.5 million people. The mean annual incidence of hip fractures for this area is 51.7 per 100,000 adults [11]. There were 343 patients (275 females; 80%) who met the inclusion criteria and had none of the exclusion criteria were enrolled in an open, prospective, and observational cohort study. All patients with acute first and second hip fractures ≥65 years who were screened for inclusion. We excluded patients with subtrochanteric, pathologic and second ipsilateral hip fractures, major concomitant injuries, multiple trauma, malignant diseases, and nonoperative treatment resulting from high surgical risk. Patients who were bed bound before fracture were also excluded from the study.

### 2.2. Materials

In order to compare the outcomes after the first and second fracture, the patients were divided into two groups. The first group included 318 (92.7%) patients who sustained their first hip fracture, while the second group included 25 (7.3%) patients who presented with a second contralateral hip fracture.

### 2.3. Procedure

At fracture presentation, information regarding sociodemographic variables (age, sex, and residential status), cognitive level, and pre-fracture physical disability were collected using a standard patient or proxy interview. Home or residential status was defined as living in own home or in an institution. During the primary hospital stay, waiting time for surgery, surgical risk, type of fracture, presence of a previous contralateral fracture, type of anesthesia, physical disability at discharge, presence of postoperative complications, and length of stay (LOS) were recorded. All assessments were performed by one tester (EDR), who was not involved in the treatment of the patients, excluding the American Society of Anesthesiologists (ASA) physical status classification of surgical risk, and the type of fracture, which were classified by the attending anesthesiologists and surgeons, respectively. Cognitive level at admission was assessed with the Short Portable Mental Level Questionnaire (SPMSQ) [12]. The 10-item questionnaire classifies the patient’s cognitive level depending on the number of correct answers as lucid (score 8–10), mild to moderate cognitive dysfunction (3–7), and severe cognitive dysfunction (0–2). In patients with an SPMSQ score less than 3, all observed variables, except for the cognitive level, were collected from a proxy. Disability is commonly defined as experiencing difficulty in carrying out activities that are essential to independent living, difficulties in performing activities of daily living (ADL), and/or instrumental activities of daily living (IADL). Pre-injury physical disability, and physical disability at discharge in terms of performing ADLs was measured using the motor subscale of the Functional Independence Measure that rates the patient independence in ranging from 1 (total assistance) to 7 (complete independence) [13]. The motor FIM scale is comprised of 13-items scored on the level of assistance required for an individual to perform activities of daily living. Physical disability 12 months after hip fracture was measured as the absolute motor FIM gain, which is the difference between 12-month follow-up motor FIM score and discharge motor FIM score. Physical disability in terms of IADL prior to fracture and at one-year follow-up was assessed using IADL scale [14]. This scale assesses eight skills, which are considered more complex than the basic activities of daily living. Persons are scored according to their highest level of functioning in that category. A summary score ranges from 0 (low function, dependent) to 8 (high function, independent). We used the ASA rating of operative risk to group patients’ physical level into one of five categories, ranging from 1 (healthy) to 5 (moribund) [15]. For the purpose of this study, ASA Classes 1 and 2 were combined, and ASA Classes 3 and 4 were combined. No patient in our study was graded as moribund. All patients with intracapsular fractures (216 patients; 62%) underwent bipolar hemiarthroplasty, whereas all patients with extracapsular fractures (128; 38%) underwent closed reduction and internal fixation with a dynamic hip screw. All patients followed a standardized postoperative rehabilitation program beginning on the first postoperative day. Early assisted ambulation with weight bearing as tolerated, and regular exercise to restore strength and mobility of the operated hip were performed two times a day for 20–30 min depending on overall postoperative health status early assisted ambulation was encouraged on the first postoperative day.

We compared the two investigated groups at fracture presentation, discharge and at one-year follow-up. At discharge, they were compared regarding motor FIM, in-hospital mortality, presence of postoperative medical complications, and LOS. Observed postoperative medical complications were delirium, pneumonia, pulmonary embolism, deep venous thrombosis (DVT), urinary tract infection (UTI), deep wound infection, pressure sores, and prosthetic dislocation. The Confusion Assessment Method was used to assess delirium on a daily basis [16]. After a year the two groups were compared regarding motor FIM and IADL and mortality. All variables were assessed by telephone interview.

The study was conducted according to the Helsinki Declaration and approved by the Ethics Commitee of School of Medicine, University of Belgrade (tracking number 440/III-8). All subjects gave written informed consent to participate in the study.

### 2.4. Statistics

Continuous variables are presented in terms of mean values with SD. Categorical values are summarized as absolute frequencies and percentages. Continuous variables between groups were tested with the *t*-test, while Chi-square test was used to test categorical variables that were expressed as numbers and percentages of patients. We used univariable and multivariable logistic and linear regression analysis with the enter method to assess the relationship between presence of second hip fracture and observed outcome variables at discharge, and one-year follow-up. In order to overcome baseline differences, the presence of a second hip fracture was adjusted on all baseline variables (age, sex, residential status, ASA, SPMSQ, preinjury motor FIM, preinjury IADL, type of fracture, type of anesthesia, and waiting time for surgery) in the multivariable regression analysis. SPSS version 21.0 was used for statistical analyses. All analyses used two-tailed significance level of *p* < 0.05.

## 3. Results

The baseline characteristics of the two examined groups are summarized in Table 1. Twenty-five (7.3%) of our patients sustained a second hip fracture after 3.00 ± 1.34 years (1–10 years) of the first hip fracture. In 72% of them the fracture was of the same type.

Patients who sustained a second hip fracture sustained more often an extracapsular fracture and had consequently more often ORIF of their extracapsular fracture performed. The only other statistically significant difference between the two groups was found regarding disability prior to fracture measured with the motor FIM score in patients with a second hip fracture. This group of patients also showed a trend towards a higher disability measured with the IADL scale, although this difference was not statistically significant (Table 1).

In hospital mortality was 7.3% (7.2% in the first hip fracture group and 8.0% in the second hip fracture patients). During the one-year follow up period after the discharge from hospital 45 (13.1%) of 343 patients were lost to follow-up, while 86 (28.8%) patients died during the first year following hip fracture after the discharge from hospital. The overall mortality has been 32.4%. (Table 2).

Univariable and multivariable regression analysis revealed no significant relationship between any observed variable and the presence of a second hip fracture at discharge from hospital (Table 2). At one-year follow-up, univariable regression analysis showed that the presence of a second hip fracture was negatively related only to IADL, i.e., patients who sustained a second hip fracture had a significantly higher IADL disability level compared to those with a first hip fracture. However, multivariable regression analysis revealed no independent relationship between the presence of a second hip fracture and any of the observed outcome variables (Table 2).

## 4. Discussion

A second hip fracture was recorded in 7.3% of our patients at average after 3.00 ± 1.34 years of the first hip fracture. In the majority of patients (72%) the fracture was of the same type. Our results are in line with Zhu et al., who also showed that most of the second hip fractures occurred in the first three years, and that the type of fracture was in most cases the same as in the first case [17]. The incidence of second hip fractures in our cohort is also in line with the results published in literature [2].

There are two important outcomes revealed in our study. The first relevant outcome is that patients with a second hip fracture showed worse functional outcome at one-year follow-up when measured with the IADL scale. The second important outcome is that patients with a second hip fracture showed no increased short-nor long-term mortality rates.

Patients with a second hip fracture performed worse regarding IADL with an average 0.16 lower IADL score compared to patients with an initial hip fracture at one-year follow-up. The importance of this finding lies in the fact that a statistically significant difference in the regression analysis regarding disability at one-year follow-up could only be observed when measured with this scale. Patients with a second hip fracture also achieved a smaller motor functional gain measured with the motor FIM scale at one-year follow up (22.37 +/− 24.08 vs. 27.08 +/−.14.43). However, univariate regression analysis revealed that the presence of a second hip fracture was not independently related to motor FIM gain at one-year follow-up.

Our results imply that the IADL scale is a more reliable measure when investigating differences in disability between patients with a second and first hip fracture compared to conventionally used ADL scales. Another finding to support this assumption is the difference in disability levels measured with the motor FIM and IADL scale prior to fracture. Our study revealed that patients with a second hip fracture had a significantly higher disability level measured with the motor FIM scale prior to fracture. Although functional disability prior to fracture measured with the IADL scale was more pronounced in patients with a second hip fracture this difference was not statistically significant. This finding confirms the fact that disabilities in IADLs precede disabilities in performing activities of daily living and loss of autonomy [18]. Thus, IADL independence is impaired in both investigated groups in our cohort prior to fracture in contrast to more impaired ADL independence in the group of patients with a secondary hip fracture.

To the best of our knowledge, this is the first study to use the IADL scale as an outcome measure in patients with a secondary hip fracture. Based on our results, we believe that IADL could be used as one of the markers of frailty and is thus a more reliable outcome measure when investigating differences in recovery from disability in patients with a second hip fracture compared to conventionally used ADL scales. Nourhashémi et al., who found a significant association between disabilities on the IADL scale and various aspects of frailty in a population of healthy elderly women, also believe that disabilities on the IADL scale could be a good tool for identifying individuals at risk of frailty among elderly persons [19].

The results of other authors concerning functional outcome after a second hip fracture are inconsistent. Two groups of authors used the New Mobility Score to assess mobility after hip fracture [1,2], and both reported no difference between patients with an initial and a second hip fracture at one-year follow-up. There are several studies who showed worse functional outcome after a second hip fracture [6,7,8,9]. Rodaro et al., who also used the FIM scale to measure functional outcome, found that this score is significantly worse within 10 to 14 days after discharge from hospital after a second fracture [6]. The findings of other authors who also reported worse mobility after the second hip fracture were limited by a small number of patients, use of non-validated measurements of walking abilities, or short-follow up period [7,8,9].

Another significant outcome of this study is that patients who sustain a second hip fracture do not have increased short- nor long-term mortality rates compared to patients with a first hip fracture. Our findings are different to those of several authors who showed that one-year death rates are greater for second hip fracture [2,3,20]. However, it remains unclear if the increased risk of death is the reflection of advanced age rather than the effect of second hip fracture. Older age and dementia are well recognized to impact survival and functional outcome after hip fracture [15,21,22]. Furthermore, older age is one of the risk factor for a second contralateral hip fracture after the initial hip fracture [22]. While the majority of authors argue that the second hip fracture per se does not increase mortality risk [2,3,5], Sobolev et al. showed that the second hip fracture increased the death above that anticipated for an increase in age in hip fracture patients [9]. A crucial difference between our results and those reported in literature is reflected in the fact that the mean age in our cohort was 78.19 +/− 7.42 years, whereas only 20% of our patients were older than 85 years. In addition, there was no statistically significant difference regarding age and cognitive status between the two groups in our cohort. In contrast, other studies reveal that the majority of both patients with a first and second hip fracture belong to the 80–89-age group [1,2], and that patients with the second hip fracture are older than patients who sustain an initial hip fracture. The lower average age of our patients compared to those reported in literature is a rational explanation for similar mortality rates in both patient groups. The younger age of patients who sustained both a first and a second hip fracture in our cohort, compared to data published in literature, is a specific epidemiologic characteristic and has a twofold explanation. First, life expectancy in Serbia is much lower than in many other European countries (77.8 for women, 72.7 for men) [23]. Second, insufficient identification of clinical fracture risk factors in the primary care setting, inadequate treatment of osteoporosis and, consequently ineffective prevention of first and second hip fractures in the geriatric population also leads to younger age of both patients groups [24].

The major strength of our study is its prospective design, and the introduction of IADL as an outcome measure for measuring disability in patients with second hip fractures. Most of the previous studies that investigated outcome after a second hip fracture, were retrospective. Additionally, we used validated outcome measures, which is not the case in all previous studies who focused on this topic. The major limitation of our study is the relatively small number of patients, which is due to the prospective design of our study. A longer duration of future prospective studies could overcome this barrier. Institutionalization as an outcome measure after one-year follow-up was also not investigated. We decided to exclude this variable after the analysis of baseline variables in our cohort, which revealed that only 2.03% patients were institutionalized prior to fracture. The decision for institutionalization in our setting is more often dependent on non-medical factors, such as cultural perception, and available financial support than on the realistic needs of the patients. Therefore, we believe that the number of patients requiring institutional care is largely underestimated in our study and cannot be meaningfully discussed. Another potential shortcoming of this study is the recall bias because the pre-injury physical disability assessments were obtained retrospectively.

Since our study revealed an association between recovery of IADL and the presence of a second hip fracture, future studies should further investigate the change of IADL during the time of recovery, and discover other variables associated with IADL disability.

## 5. Conclusions

In summary, our study reveals that the presence of a second hip fracture is an independent predictor of higher disability levels measured with the IADL scale at one-year follow-up. We believe that disabilities on the IADL scale are a marker of frailty, and that this scale is thus a more reliable outcome measure when investigating functional outcome in patients with a second hip fracture compared to conventionally used ADL scales

In addition, we believe that patients similar mortality rates in patients with a first and second hip fracture are due to the general younger age of patients in our cohort compared to data in literature. This implies that outcome varies between different settings due to dissimilar and specific epidemiologic characteristics of elderly patients sustaining non-simultaneous bilateral hip fractures. Our results require confirmation by future longitudinal studies on a larger group of patients.

## Figures and Tables

**Table 1 geriatrics-05-00067-t001:** Baseline characteristics of study patients.

	First Hip Fracture	Second Hip Fracture	*p* Value
Number of patients, f (%)	318 (92.7%)	25 (7.3%)	
Gender ^‡^ Male Female	66 (20.8%) 252 (79.2%)	2 (8.0%) 23 (92.0%)	0.124
Age (y) ^†^	78.10 ± 7.53	78.96 ± 6.02	0.578
Residential status ^‡^ Home Institution	311 (98.4%) 5 (1.6%)	23 (92.0%) 2 (8.0%)	0.087
SPMSQ ^†^	7.30 ± 2.94	6.54 ± 3.12	0.225
Operative risk ^‡^ ASA 1,2 ASA 3,4	185 (58.4%) 132 (41.6%)	15 (60.0%) 10 (40.0%)	0.873
Motor FIM preoperative ^†^	85.34 ± 10.36	80.92 ± 13.06	0.045
IADL preop ^†^	5.47 ± 2.72	4.48 ± 2.68	0.086
Type of fracture ^‡^ Intracapsular Extracapsular	206 (64.8%) 112 (35.2%)	9 (36.0%) 16 (64.0%)	0.004
Waiting time for surgery ^‡^ <48 h ≥48 h	33 (10.4) 285 (89.6%)	3 (12.0%) 22 (88.0%)	0.799
Type of anesthesia ^‡^ Local General	155 (49.7%) 157 (50.3%)	14 (56.0%) 11 (44.0%)	0.543
Type of surgery ^‡^ Hemiarthroplasty Dynamic Hip Screw	206 (64.8%) 112 (35.2%)	9 (36.0%) 16 (64.0%)	0.004

^†^ Values given as mean with SD (mean ± SD), or as median with interquartile range depending on the normality of distribution; ^‡^ values given as number of patients with percentage in parentheses; ASA = American Society of Anesthesiologists; SPMSQ = Short Portable Mental Level Questionnaire; IADL = Instrumental activities of daily life; FIM = Functional Independence Measure.

**Table 2 geriatrics-05-00067-t002:** Outcomes up to one year and second hip fracture as theirs predictor.

Variable	First Hip Fracture (*n* = 318)	Second Hip Fracture (*n* = 25)	OR (95% CI)/ b (95% CI) ^1^	*p*-Value Unadjusted ^2^	OR (95% CI)/ b (95% CI) ^1^	*p*-Value Adjusted ^3^
Discharge						
Death; dead/alive, %	23/294 (7.3%)	2/23 (8.0%)	1.12 (0.25–5.03)	0.887	1.12 (0.19–6.43)	0.896
Complications	138/179 (43.5%)	11/14 (44.0%)	1.01 (0.45–2.29)	0.978	0.83 (0.34–2.06)	0.693
FIM	42.14 ± 17.82	42.64 ± 15.70	0.01 (−6.65 to 7.82)	0.875	0.07 (−0.87 to 9.67)	0.101
Length of hospital stay	31.05 ± 17.53	33.96 ± 22.08	0.04 (−4.39 to 10.22)	0.434	0.03 (−5.64 to 9.81)	0.569
From discharge to one year						
Death; dead/alive, %	80/196 (29.0%)	6/16 (27.3%)	0.92 (0.35–2.43)	0.865	0.68 (0.21–2.23)	0.527
One year						
Death; dead/alive+lost to follow-up; %	80/238 (33.6%)	8/19 (42.10%)	1.15 (0.43–3.06)	0.782	0.86 (0.26–2.82)	0.804
Motor FIM gain ^4^	27.08 ± 14.43	22.37 ± 14.08	−0.04 (−10.66 to 5.77)	0.558	−0.10 (−14.28 to 1.89)	0.133
IADL ^4^	4.43 ± 2.82	2.69 ± 2.44	−0.16 (−3.17 to −1.40)	0.018	0.03 (−0.73 to 1.40)	0.532

^1^ Logistic regression analysis with odds ratio and its 95% CI or Linear regression analysis with partial coefficient of regression and its 95% CI where appropriate. ^2^ With the presence of a second hip fracture as independent variable in univariate regressions models. ^3^ With the presence of a second hip fracture as independent variable adjusted on all baseline variables in multivariate regressions models. ^4^ Calculated on 196 first hip fracture patients and 16 s hip fracture patients. From 343 patients 45 patients were lost to 1-year follow-up 42 in 1st hip fracture group and 3 in 2nd hip fracture group) and 25 died in hospital (23 dead with1st hip fracture and 2 dead with 2nd hip fracture).

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
