# Peer review of "Instrumental Activities of Daily Living—A Good Tool to Prospectively Assess Disability after a Second Contralateral Hip Fracture?"

_geriatrics, 2020, doi:10.3390/geriatrics5040067_

Round 1

Reviewer 1 Report

Good overall design and research question on function following 2 Nd hip fracture. Good study design and outcome measures between 1 st and 2nd hip fractures. 
conclusion of Iadl as important finding of study.
Need to improve English spell check. 

Author Response

Thank you so much for your suggestions. With the spell check I corrected several English mistakes. If there is something else to correct, I will be happy to do so.

Reviewer 2 Report

In this paper, the authors aim to determine the outcome for patients who sustain a second hip fracture compared with those who sustain a first fracture, and to define the optimal measure to evaluate functional outcome after second hip fracture. The paper is easy to read and insightful. However, there are some open topics that I think would improve the quality of the paper.

Introduction

The introduction should provide an overview of the state of the art of the topic, what has already been done and what is the aim of the paper. In my opinion, this part should be implemented with a more in-depth search of the literature.

Materials and Methods

This session is well written even if it should be organized in a more schematic way to make it more readable, highlighting the inclusion and exclusion criteria followed in the selection of subjects.

Moreover, if it is possible, I suggest including a description of the rehabilitation treatment used.

Discussion

At the end of the Discussion session, I suggest adding a paragraph on possible future studies, such as investigating the rehabilitation protocols currently used in hip fractures, by inserting the following reference [Maranesi, E., Riccardi, G.R., Lattanzio, F., Di Rosa, M., Luzi, R., Casoni, E., Rinaldi, N., Baldoni, R., Di Donna, V., Bevilacqua, R. Randomised controlled trial assessing the effect of a technology-assisted gait and balance training on mobility in older people after hip fracture: Study protocol (2020) BMJ Open, 10 (6), art. no. E035508]

Valid for the entire manuscript: review the spelling of words. For example, line 14 deteremine (determine).

Author Response

Thank you for your suggestion.

  1. Introduction : We changed the introduction according to your suggestions (lines 41-43) adding more specific information related to existing available data. A new search of the literature did not reveal any new studies on this topic that were not already cited.
  2. Materials and methods: This session was organized better by adding new subtitles. Inclusion and exclusion criteria were formulated more precisely (lines 57-61).  The rehab protocol was described more in debth (lines 100-104)
  3. Discussions: We already highligh the need for future studies. However, the study you propose us to cite is not related to the topic of our investigation, and we firmly believe that it does not fit in.